# Comparisons between the ON- and OFF-edge motion pathways in the *Drosophila* brain

Kazunori Shinomiya[1]*, Gary Huang[1], Zhiyuan Lu[1,2], Toufiq Parag[1,3], C Shan Xu[1], Roxanne Aniceto[1], Namra Ansari[1], Natasha Cheatham[1], Shirley Lauchie[1], Erika Neace[1], Omotara Ogundeyi[1], Christopher Ordish[1], David Peel[1], Aya Shinomiya[1], Claire Smith[1], Satoko Takemura[1], Iris Talebi[1], Patricia K Rivlin[1], Aljoscha Nern[1], Louis K Scheffer[1], Stephen M Plaza[1], Ian A Meinertzhagen[2]*

[1]Janelia Research Campus, Howard Hughes Medical Institute, Ashburn, United States; [2]Department of Psychology and Neuroscience, Dalhousie University, Halifax, Canada; [3]School of Engineering and Applied Sciences, Harvard University, Cambridge, United States

**Abstract** Understanding the circuit mechanisms behind motion detection is a long-standing question in visual neuroscience. In *Drosophila melanogaster*, recently discovered synapse-level connectomes in the optic lobe, particularly in ON-pathway (T4) receptive-field circuits, in concert with physiological studies, suggest a motion model that is increasingly intricate when compared with the ubiquitous Hassenstein-Reichardt model. By contrast, our knowledge of OFF-pathway (T5) has been incomplete. Here, we present a conclusive and comprehensive connectome that, for the first time, integrates detailed connectivity information for inputs to both the T4 and T5 pathways in a single EM dataset covering the entire optic lobe. With novel reconstruction methods using automated synapse prediction suited to such a large connectome, we successfully corroborate previous findings in the T4 pathway and comprehensively identify inputs and receptive fields for T5. Although the two pathways are probably evolutionarily linked and exhibit many similarities, we uncover interesting differences and interactions that may underlie their distinct functional properties.

DOI: https://doi.org/10.7554/eLife.40025.001

*For correspondence:
shinomiyak@janelia.hhmi.org (KS);
I.A.Meinertzhagen@Dal.Ca (IAM)

Competing interests: The authors declare that no competing interests exist.

## Introduction

Over half a century ago, *Hassenstein and Reichardt (1956)* working on the beetle *Chlorophanus*, and later *Reichardt (1961)* working on flies and *Barlow and Levick (1965)* studying rabbit retinal ganglion cells, all independently presented evidence for motion detection circuits that incorporate a delay-and-compare strategy. In both insect and mammalian model groups, two or more independent, parallel inputs from upstream neurons provide input to elementary motion detector (EMD) circuits. Both models use a similar mechanism to compute the direction of motion, but they differ depending on how they produce a direction-selective response. The Barlow-Levick type circuit detects the preferred-direction signals by suppressing signals in the non-preferred direction; the Hassenstein-Reichardt detector generates an enhancement of signals in the preferred direction (*Borst and Helmstaedter, 2015*).

The fly's optic lobe consists of four consecutive neuropils: the lamina, medulla, lobula, and lobula plate. Each of these comprises columnar units that correspond to the array of ommatidia in the retina. The motion pathway in the optic lobe arises from the photoreceptor cells (PRs), which receive light signals in the compound eye and extend their axons to the lamina. R1–R6 cells expressing

rhodopsin rh1 provide signals to lamina monopolar cells in the lamina cartridges, which project to the distal medulla. The lamina neurons are presynaptic to various types of medulla neurons in the distal medulla. Among them, medulla columnar neurons including Mi, Tm, and TmY cells further provide inputs to the dendritic arbors of T4 in the M10 layer of the medulla and T5 in the Lo1 layer of the lobula.

The dendritic arbors of T4 cells receive parallel inputs from multiple columns, and a single arbor receives inputs from columns that signal different positions of the visual field, depending on the cell types of the input neurons (*Takemura et al., 2017*). Recent developments in techniques for three-dimensional electron microscopy (3D-EM) have accelerated the identification of neurons and their synaptic circuits, or their connectome, in the brain of the fruit fly *Drosophila melanogaster* (*Meinertzhagen, 2016*; *Meinertzhagen, 2018*). In the visual system, motion detection pathways in the optic lobe have been a prominent goal for such connectomic approaches, which identify the component neurons using 3D-EM reconstructions of their arbors.

The medulla dendritic arbors of T4 cells provide a substrate for the elementary motion detector (EMD) in the ON-edge motion pathway (*Borst, 2014*; *Joesch et al., 2010*; *Maisak et al., 2013*). Using serial-section transmission EM (ssTEM), *Takemura et al. (2013)* identified Mi1 and Tm3 as major inputs to the T4 cell dendrites. A later approach using focused ion beam scanning EM (FIB-SEM) (*Takemura et al., 2017*) comprehensively revealed other medulla neurons providing inputs to T4. These medulla neurons relay input to T4 from L1, the first of two repeated neuron classes in the first neuropil, or lamina; L1 in turn receives input from the terminals of photoreceptors R1–R6 in the overlying compound eye (*Meinertzhagen and O'Neil, 1991*; *Rivera-Alba et al., 2011*).

Complementary to the T4 cells, narrow-field T5 cells constitute the first output stage of the OFF-edge pathway (*Borst, 2014*; *Joesch et al., 2010*; *Maisak et al., 2013*), and some of T5's input neurons have also been identified from their terminals reconstructed using ssTEM (*Shinomiya et al., 2014*). These inputs relay signals from L2 cells, which partner L1 in all columns, or cartridges, of the lamina and which also receive input from R1–R6 (*Meinertzhagen and Sorra, 2001*; *Meinertzhagen and O'Neil, 1991*; *Rivera-Alba et al., 2011*). Therefore, the separation between the ON and OFF motion pathways is already established at the level of the lamina neurons.

Finally, T4 and T5 cell axons transfer motion information to the fourth neuropil, or lobula plate, where it is integrated and further processed to extract specific motion modalities, before being conducted to the central brain by visual projection neurons (VPNs). VPNs include various types of lobula plate tangential neurons (LPTCs) and lobula plate/lobula columnar cells (*Klapoetke et al., 2017*; *Mauss et al., 2015*; *Scott et al., 2002*).

The ON and OFF motion pathways are similar in their function, component neurons, and patterns of synaptic connections. Both T4 and T5 cells are direction-selective neurons, and each is further grouped into four subtypes: T4 as T4a, T4b, T4c and T4d; and T5 as T5a, T5b, T5c, and T5d. These T4 and T5 cells specifically signal motion in the four canonical directions. The subtypes a–d detect front-to-back, back-to-front, upward, and downward motion, respectively (*Maisak et al., 2013*). Each subtype projects its axon to one of the lobula plate's four strata (*Fischbach and Dittrich, 1989*), depending on the direction of motion that it signals (*Maisak et al., 2013*). Developmentally, both T4 and T5 are known to originate from the same subset of progenitor cells in the inner proliferation center and to express a proneural gene, Atonal, uniformly (*Apitz and Salecker, 2016*; *Oliva et al., 2014*).

Given the dimensional constraints of the respective ssTEM and FIB-SEM datasets, however, the T4 and T5 pathways, and their respective input neurons, have been reconstructed independently in separate reports using 3D-EM methods. Series of ultrathin sections have been used to identify medulla cell inputs to T4 cells; these included medulla intrinsic (Mi) and transmedulla (Tm) cells but not their terminals in the lobula, which were lacking from the EM dataset (*Takemura et al., 2013*). Similarly, inputs to T5 terminals in the lobula arise from Tm cells, but the medulla arbors of these were also lacking from previous reconstructions (*Shinomiya et al., 2014*). Subsequent reports that repeated the analysis of cells for seven medulla columns, using FIB-SEM (*Takemura et al., 2015*), also failed to identify the lobula, but comprehensively identified additional inputs to, and connections between, T4 cells (*Takemura et al., 2017*). Consequently, results from these studies cannot be compared directly to the same field size and at the same resolution in a single dataset. This makes it difficult to recognize and resolve deep similarities in the inputs to both pathways, which might support further evolutionary comparisons between those inputs (*Shinomiya et al., 2015*), and

which might also enable functional comparisons, especially for the inputs to T5 which to date are known only for four main Tm cells: Tm1, Tm2, Tm4, and Tm9.

In this study, we exhaustively identified the synaptic inputs to T5 cells, and described their spatial layouts. We also assessed the anatomical properties of the dendritic terminals of T4 and T5, after identifying all neurons that have synaptic contacts with the motion-sensing output cells in the medulla and lobula. As a result, our report concludes the connectomic analysis of both the ON- and the OFF- motion-sensitive pathways in *Drosophila*.

## Results

### Identifying the synaptic partners of T4 and T5 cells

The FIB-SEM dataset used in this study covers the entire depth of the medulla, lobula, and lobula plate, as well as the proximal depth of the lamina (*Figure 1A*), and is large enough to include the motion pathways from the lamina cell inputs, through the medulla, to T4 and T5 cells in the lobula

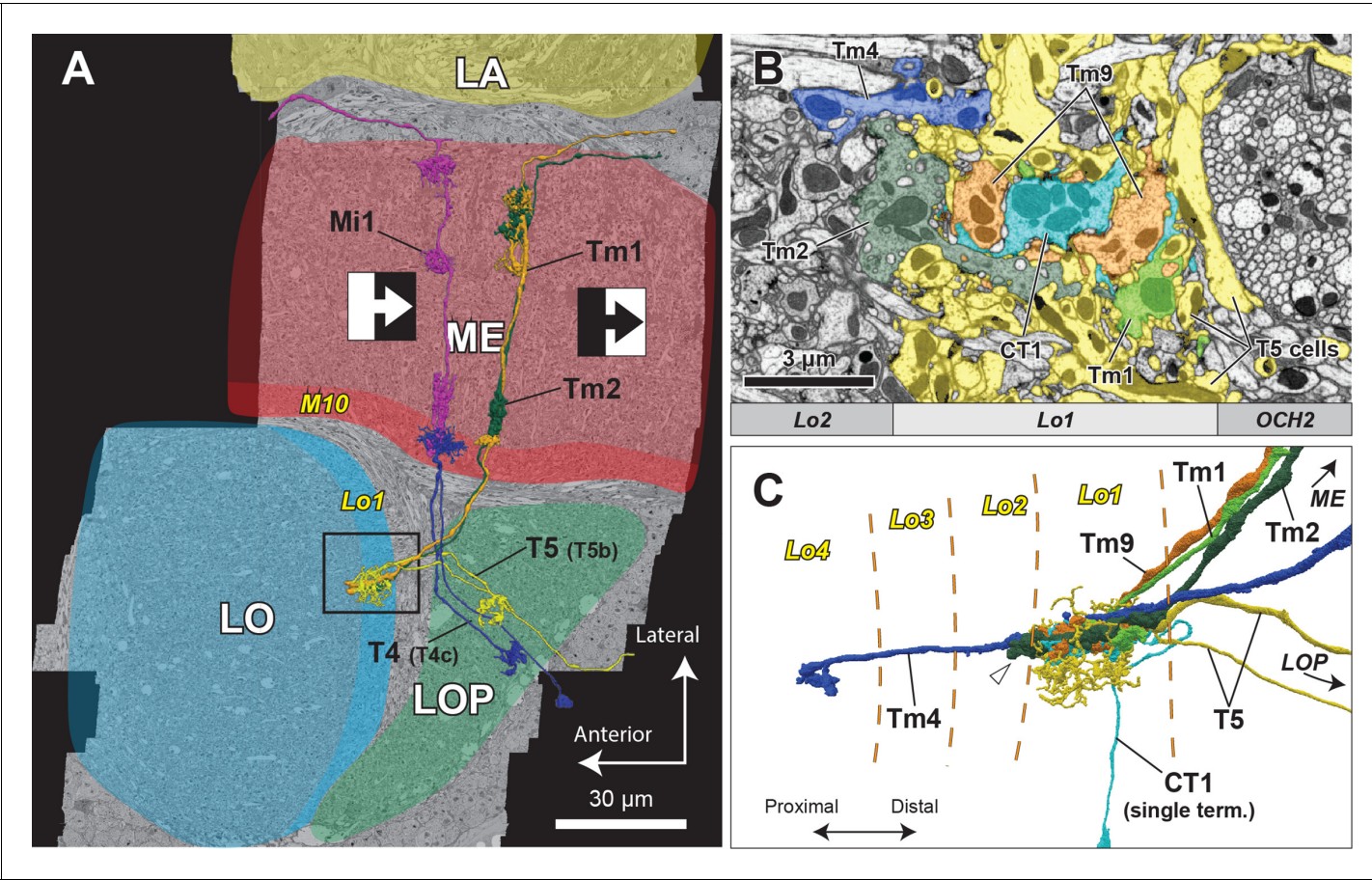

**Figure 1.** Horizontal section of the four optic lobe neuropils and their reconstructed neurons. (**A**) In the ON-edge pathway, the T4 cells receive inputs at the tenth layer of the medulla (M10) from upstream cells, whereas in the OFF-edge pathway, the T5 cells receive inputs at the first layer of the lobula (Lo1), before projecting to the lobula plate. Here, in a horizontal re-sliced section of the dataset, Mi1 and Tm1/Tm2 cells are shown as representative input neurons to the T4 and T5 cells. LA, lamina; ME, medulla; LO, lobula; LOP, lobula plate. (**B**) Axon terminals of Tm cells and T5 cells in Lo1. The area roughly corresponds to the boxed inset in (**A**). T5 cell dendrites (in yellow) demarcate the Lo1 layer. Terminals of Tm1, Tm2, Tm9, and CT1 neurons form packed columnar units and have extensive synaptic contacts with T5 cells. Tm4 cells form no part of these columnar units but run between the columns, and also have synapses with T5 cells in Lo1. (**C**) 3D reconstruction of the neurons. One representative T5 cell (yellow) is shown along with Tm cells and a CT1 terminal. The proximal tip of the axon terminal of Tm2 (arrowhead) extends slightly beyond the boundary between Lo2 and Lo1. Tm4 extends its axon further and terminates in Lo4 (*Fischbach and Dittrich, 1989*).

DOI: https://doi.org/10.7554/eLife.40025.002

plate. We have identified a complete subset of neurons that form either pre- or postsynaptic partnerships with the dendritic arbors of both T4 and T5 cells, using the numbers and locations of presynaptic contacts, each identified by a T-bar presynaptic ribbon and multiple postsynaptic dendrites (*Takemura et al., 2008*) (see Materials and methods for detailed processes of neuron tracing). Examples of reconstructed cells, including those of T5 and some of its synaptic partners in stratum Lo1, are shown in *Figure 1B and C*. The dendritic arbors of T5, as well as most of the input neuron terminals, are confined to Lo1. Tm1, Tm2, and Tm9 have small columnar terminals which look similar to each other. The terminals of Tm1 and Tm9 are entirely within stratum Lo1, which is defined as the stratum innervated by the T5 cell dendrites (*Fischbach and Dittrich, 1989*), whereas Tm2's terminal extended into stratum Lo2, slightly deeper than Lo1 (*Figure 1B and C*).

On the basis of reconstructed neurons and their synapses, we comprehensively identified input and output neurons of T4 and T5 in the medulla and lobula (*Figure 2*). The TmY18 cell, which receives inputs from T4 but which was not reported in *Takemura et al. (2017)*, is also included here. TmY18 is postsynaptic only to T4 in M10, and lacks synaptic contacts with T5. For the T5 cells, in addition to the four types of Tm cells reported previously (*Shinomiya et al., 2014*), four additional types of neurons were identified presynaptic to the dendrites. Neurons that have synaptic contacts with the T5 cell terminals in Lo1 are shown in *Figure 3*. CT1, a large tangential, GAD1-immunopositive (*Takemura et al., 2017*) element, was presynaptic to the T5 dendrites. Its terminals in the medulla are reported to contact T4 cell neurites (*Takemura et al., 2017*). CT1 spans the opposing faces of M10 and Lo1 to form modular associations, that is one per column, with both T4 and T5 populations in both the medulla and lobula (*Figure 3B and C*). The dataset from the present study reveals that these columnar terminals of CT1 (*Figure 3D*) are both pre- and postsynaptic to the neurites of T4 and T5. In particular, the terminals in Lo1 are physically larger than those in M10, and the input synapses from CT1 to T5 are actually more numerous than those to T4, making CT1 one of the prominent input elements to the T5 cell. Each terminal of CT1 is tightly packed with the terminals of Tm1, Tm2, and Tm9, and forms regularly aligned columnar structures embedded in Lo1 (*Figure 1B*).

The wide-field neurites of CT1 are, in several respects, a novel player among the input neurons to T4 and T5. First, they form repeated synaptic modules in each neuropil column of the medulla and lobula (*Figure 1C*), where they provide input to the dendrites of T4 and T5, at about 15 synapses per column to T4 and about 60 to T5 (essentially equal numbers to subtypes a-d). Second, besides being presynaptic to the T5 cell dendrites, CT1 also receives inputs from Tm1 and Tm9 cells in Lo1 (7.0 ± 3.1 and 23.3 ± 2.6 synapses per CT1 terminal, respectively (n = 7)), and provides outputs to Tm9 (14.9 ± 3.2 synapses per CT1 terminal (n = 7)). Third, that input is GAD1 immunopositive (*Takemura et al., 2017*) and therefore putatively inhibitory. Fourth, the modules are connected by long neurites that stretch across stratum M10 of the medulla and stratum Lo1 of the lobula, and also between these neuropils, from a soma in the central brain (*Takemura et al., 2017*). Fifth, the connecting neurites are ~0.1 µm in diameter, in contrast to 0.5 ~ 1 µm in common Mi or Tm cell axons, and their average length (44 µm) is much longer than that required to connect between columns (~5 µm). We note that this need not be a general feature of lobula tangential neurons, and is not shown for example by LT33 (*Figure 3C–E*).

Another cell type, TmY15 (*Takemura et al., 2017*), which has neurites in the medulla, lobula, and lobula plate and is also GAD1 positive (*Takemura et al., 2017*), is presynaptic to both the T4 and T5 cells in M10 and Lo1, respectively. Like CT1, this cell type is putatively inhibitory. It extends over areas of the medulla and lobula wider than ten columns, and the innervating areas in the two neuropils roughly overlap each other retinotopically. TmY15 does not receive direct inputs from lamina cells in the medulla but it receives inputs from several types of Mi, Tm, and TmY cells, most notably from Tm3 and Tm4 cells, at its arbors in both the medulla and lobula. In the lobula plate, the cell has input synapses predominantly in the second and third strata, where it receives inputs from the axon terminals of the T4b/T5b and T4c/T5c cells, respectively.

We also found that two types of Lo1-specific wide-field cells form synaptic inputs to the T5 terminals (*Figure 3E–I*). One of these has a wide tangential arbor covering the entire Lo1 stratum, and we name it LT (lobula tangential) 33, previously reported cells of the same class being LT1–LT10 (*Fischbach and Dittrich, 1989*), and LT11–LT12, and LT31–LT32 (*Otsuna and Ito, 2006*). There is only a single LT33 cell in each hemisphere, and the cell body is located in the central brain, near the midline (*Figure 3E and F*), which is also near the cell body site of CT1 (*Shinomiya et al., 2015*; *Takemura et al., 2017*). LT33 also has some branches in the neuropil of the posterior lateral

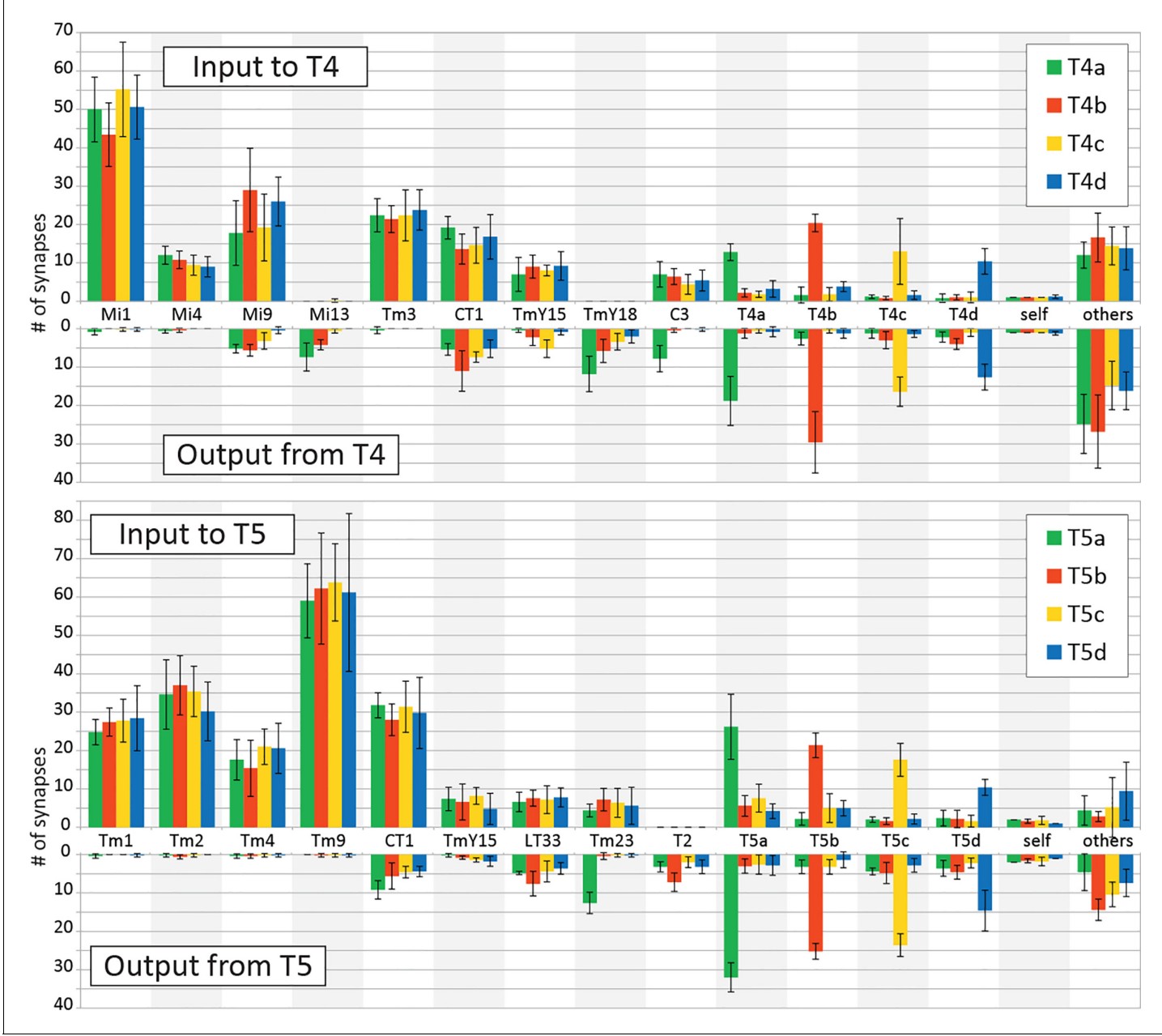

**Figure 2.** Input and output synapses of T4 and T5 cells in the medulla and lobula. Histograms indicating the average numbers of synapses per connection for a single T4 or T5 cell, mean (+SD) for five representative cells selected and completely traced for each T4/T5 subtype. Only connections with ≥ 3 contacts are independently shown in the graphs; weaker connections are combined in 'others'. All synaptic sites were manually verified, and their pre- and postsynaptic partners were then also exhaustively identified.

DOI: https://doi.org/10.7554/eLife.40025.003

protocerebrum (PLP). The other type of cell is identified as a previously described neuron, Tm23, with a cell body in the medulla cortex, distal to the medulla neuropil, which after penetrating the medulla projects an axon to stratum Lo1 (*Figure 3G–I*) (*Fischbach and Dittrich, 1989*). Tm23 seems to have no synapses in the medulla, and the projecting arbor of a single cell in the lobula covers an area of roughly 30 × 40 μm. Both LT33 and Tm23 are postsynaptic at inputs from T5, Tm23 being postsynaptic only to cell T5a in Lo1.

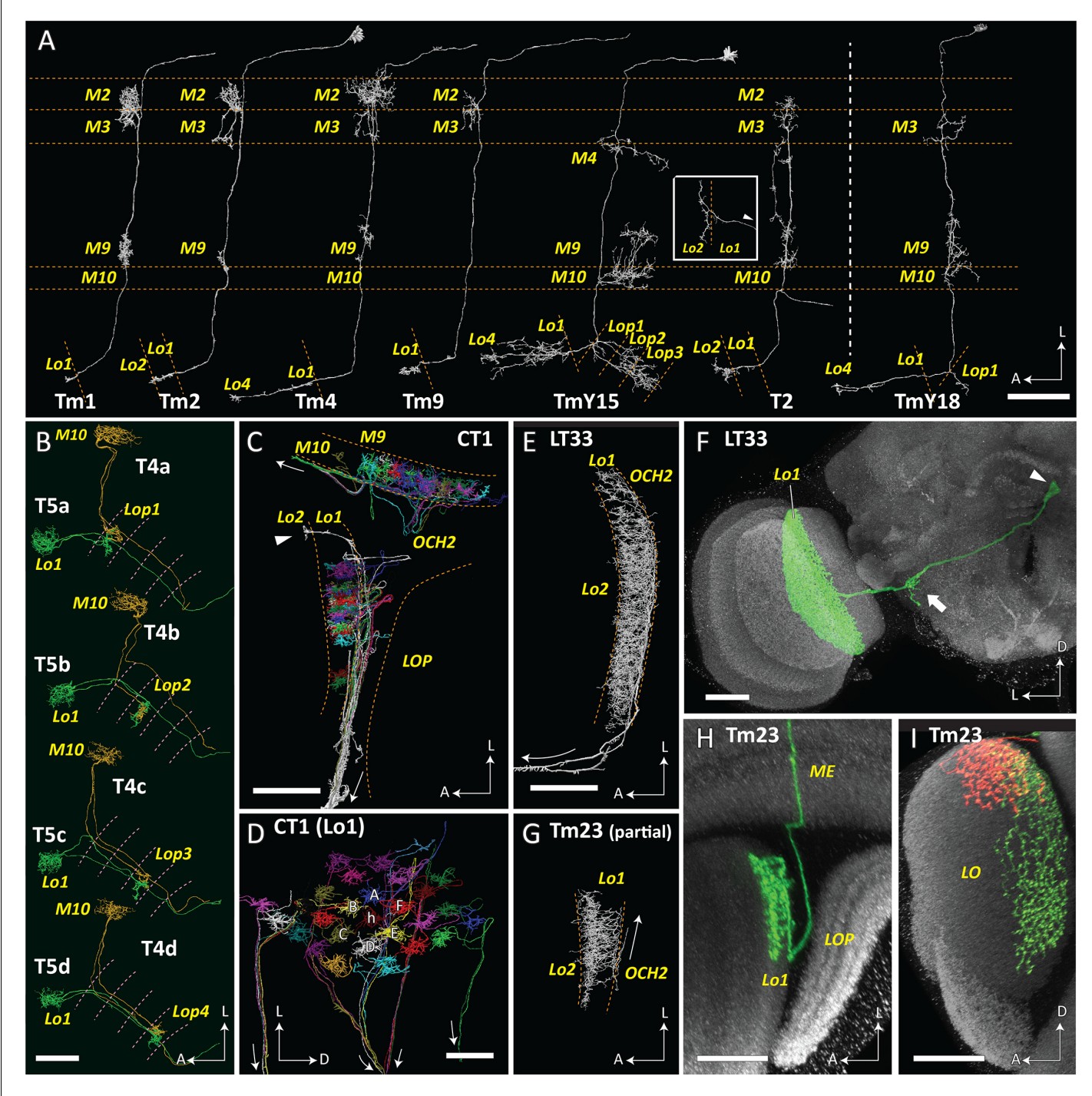

**Figure 3.** T4, T5 and their synaptic partner neurons. (**A**) T5 forms synaptic contacts with Tm1, 2, 4, 9, TmY15 and T2 cells. T4 is presynaptic to a newly identified TmY18 cell in M10. These synaptic partners of T5 and T4 have cell bodies in the medulla cortex distal to stratum M1, except T2 has a soma between the medulla and lobula. Tm and T2 cells receive inputs mainly in the medulla and project to the lobula, whereas TmY cells project to both the lobula and the lobula plate. All reconstructed cell profiles closely capture the profiles of those reported from Golgi impregnation (*Fischbach and Dittrich, 1989*) except the lobula terminal of T2 (inset), shown here from a different angle (lateral view, arrowhead indicates the outgoing fiber connected to the medulla and cell body). (**B**) Paired T4 and T5 cells. T4/5a, b, c, and d cell pairs project to the lobula plate, to strata Lop1, 2, 3 and 4, respectively. Their cell bodies are located in the lobula plate cortex, posterior to the lobula plate. (**C**, **D**) Parts of CT1 reconstructed in both the medulla and lobula reveal the morphology of its color-coded columnar terminals. Fibers connecting the terminals enter the second optic chiasm (OCH2) before exiting the optic neuropils (trajectories shown by arrows). Most branches terminate in medulla stratum M10 or lobula stratum Lo1, but a few terminals project deeper, into Lo2 (arrowhead in (**C**)). Columnar terminals in Lo1 corresponding to a home (h) column and the surrounding six columns (A–F) are

*Figure 3 continued on next page*

*Figure 3 continued*

indicated in (D). (E) A partial reconstruction of LT33, a novel neuron in lobula stratum Lo1. (F) Optic lobe and central brain projection pattern of an LT33 cell visualized using multicolor stochastic labeling (MCFO; see Materials and methods). The arrowhead indicates the cell body, and the arrow shows the site of branching in the posterior lateral protocerebrum. (G) Partial reconstruction of Tm23 in stratum Lo1. (H, I) Tm23 cells and their projection patterns in the optic lobe visualized using MCFO. Images show two different views generated from the same confocal stack. Images in (F), (H), and (I) show resampled views generated from confocal stacks using Vaa3D. For clarity, the images in (F) and (H) were segmented to omit additional labeled cells (distinguished by labeling color and position) that are also present in the displayed volume. In (I), red and green signals each represent single Tm23 cells. Scale bars: (A) 20 μm; (B), (C), (E), (G) 10 μm; (D) 5 μm; (F), (H), (I) 20 μm.
DOI: https://doi.org/10.7554/eLife.40025.004

The T2 cell is postsynaptic at a few contacts from each of the four T5 subtypes in the lobula (*Figure 2*). T2 has its main terminal in the Lo2 stratum, but it touches the tips of the T5 cell dendritic arbors on its distal side and there forms synaptic contacts (*Figure 3A*).

The present dataset also reveals some novel minority connections that have a few synaptic contacts. Two types of neurons, Mi13 and TmY18, receive input from T4 cells but lack direct output back to T4 (*Figures 2* and *3A*). Interestingly, Mi13 is selectively postsynaptic to T4a (at $7.4 \pm 3.6$ synapses) and T4b (at $4.2 \pm 1.3$ synapses), whereas T4c ($0.6 \pm 0.5$ synapses) and T4d (zero synapses) have fewer or no such synapses. In addition, C3 is postsynaptic specifically to T4a ($7.8 \pm 3.4$ synapses) but not to other T4 subtypes. C3 is widely reported to be GABAergic (*Kolodziejczyk et al., 2008*; *Sinakevitch et al., 2003*), suggesting that C3 inhibits the detection or processing of motion by suppressing all subtypes of T4 cells when it receives front-to-back signals from T4a. This connectivity pattern is comparable to that of Tm23 and T5 cells. Thus although the morphologies of C3 and Tm23 are quite different, they have similar connections with the T4 and T5 cell arbors, respectively.

## Spatial displacements of synaptic inputs to T4 and T5 cells

To identify candidate inputs to an EMD, we mapped the synaptic sites distributed over the dendrites of both T4 and T5 cells and compared these maps with plots for T4 made by *Takemura et al. (2017)*, using T4c and T5c as representatives of both cells (*Figure 4*).

In T4, synapses from all of the neurons were distributed in the manner previously reported (*Takemura et al., 2017*). In the case of T5, input synapses from Tm1, Tm2, Tm4, and TmY15 were distributed specifically at the middle region of the dendrites, in a pattern similar to that for inputs from Mi1, Tm3, and TmY15 to T4. Synaptic inputs from Tm9, on the other hand, localized to the tip and CT1 to the base of the dendritic arbors, similar to the distribution patterns of Mi9 and Mi4/CT1 synapses on T4 dendrites. Input synapses from LT33 and Tm23 were distributed at the tips to the middle of the dendritic arbors (*Figure 4*). The dendritic arbors of T4/T5 cells have presynaptic organelles ('T-bars') in addition to postsynaptic sites. We plotted the distribution of T-bar sites, which were restricted to the middle regions of the dendritic arbors in both T4 and T5 (*Figure 4*). With few exceptions, connections between T5 neurites occur between the same subtypes, for example from T5c to T5c (*Figure 4*), consistent with the previous report (*Takemura et al., 2017*) of subtype specificity among T4 cells. Like T4, T5 receives inputs from T5 of the same subtype localized to their dendritic tips.

Among the neurons providing inputs to T4 and T5 cell dendrites (*Figure 4*), TmY15, LT33, and Tm23 lack clear columnar structures in the lobula or medulla. It is therefore unlikely that they can contribute to the function of a classical EMD circuit, which requires multiple parallel inputs from upstream modular neurons. We therefore plotted the distribution ranges of the columnar cell terminals — namely Mi1, Mi4, Mi9, Tm3, C3, and CT1 terminals for T4; and Tm1, Tm2, Tm4, Tm9, and CT1 terminals for T5 (*Figure 5*). For the Mi and Tm cell inputs to T4 and T5, the inputs from lamina cells (L1, L2, and L3) were omitted. Although Tm3 and Tm4 receive inputs from more than one column in the medulla, there is no evidence of spatial bias between the input columns and the location of Tm3 or Tm4 cells, and the contribution of Tm3 and Tm4 to the EMD circuit is relatively small compared with those of other single-column cells, such as Mi1, Tm1, Tm2, and Tm9 (*Takemura et al., 2008*; *Takemura et al., 2017*). We therefore conclude that the distribution of the input sites on the T4 and T5 dendrites reflects positions corresponding to their visual receptive fields.

The spatial distributions of inputs from these neurons to T4 and T5 were calculated by annotating the neurons according to the columns they occupy in the medulla and lobula. The shape of the

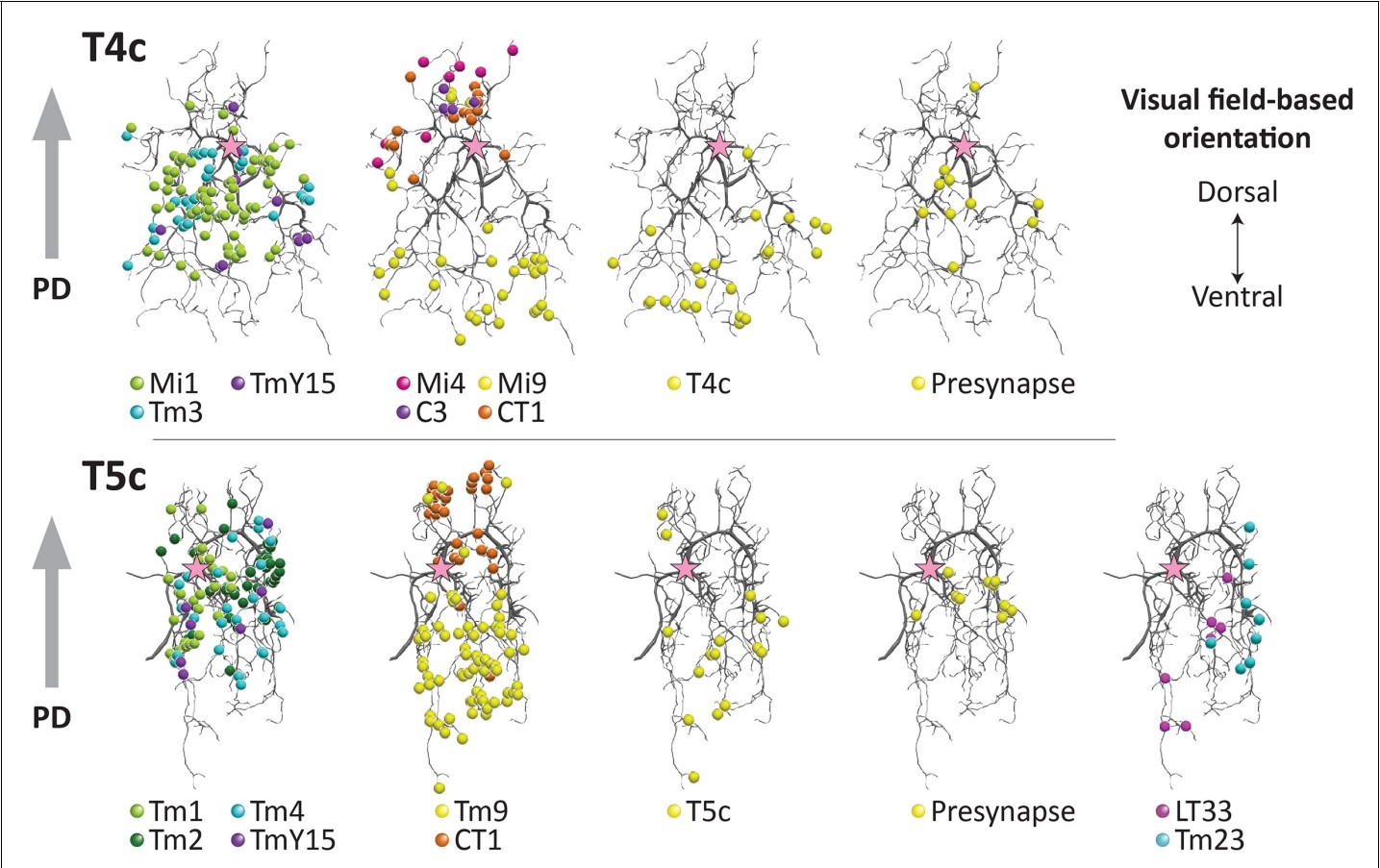

**Figure 4.** Distribution of synaptic sites over T4 and T5 dendritic arbors. Proofread synapses are plotted on the dendritic arbors of a T4c (top panels) and T5c (bottom panels) cell. Puncta are postsynaptic sites (inputs to the T4/T5 cells) except those shown as 'Presynapse', which illustrate the positions of T-bar, output sites. Both T4c and T5c detect upward motion, and other subtypes of T4 and T5 cells demonstrate similar distribution patterns(not shown). Pink stars indicate the first branch point of the arbor. PD, preferred direction.

DOI: https://doi.org/10.7554/eLife.40025.005

columns and the array formed by their cross-sections differ in different strata and according to their location in the visual field. The array of columns is considerably compressed in the A–P axis in M10 and Lo1 when compared with the hexagonal array of the compound eye and the corresponding array of underlying lamina cartridges (*Figure 5A–D*), so the projected visual field is distorted in these strata despite the conservation of their retinotopy. For analysis, we therefore approximated the column arrays in M10 and Lo1 to a regular hexagonal array. When the spatial displacements of the different types of input neurons were calculated using information from the distribution of proofread synapses (*Figure 5E*, left two columns),their spatial displacements became even clearer, so that inputs from different neuron types were obviously clustered into three groups in each of T4 and T5, with the spaces between the centers of the clusters roughly corresponding to the distance between neighboring columns.

## Evaluation of automated synapse prediction

To identify input partners for T4 and T5, in addition to tracing and identifying the neurons themselves, we also needed to annotate their synaptic sites rapidly, accurately and comprehensively. For this task, we initially identified both pre- and postsynaptic sites by recruiting an automated synapse prediction algorithm, which annotated hundreds of thousands of synaptic sites efficiently. To evaluate the quality of the prediction algorithm, and to check if such predicted synapses could be used for connectivity and functional analyses, we then exhaustively checked the predicted input and

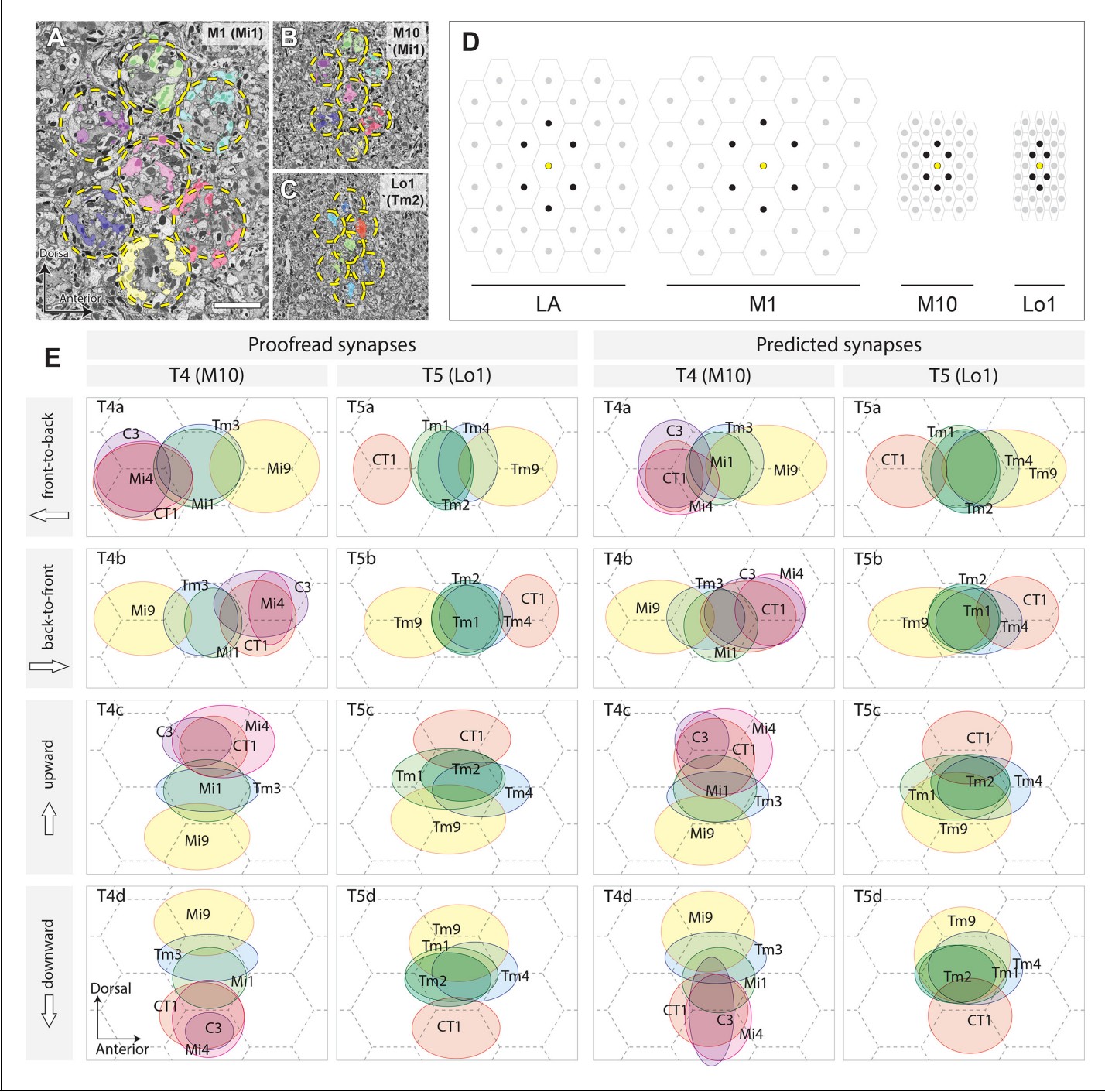

**Figure 5.** Columnar units in the optic neuropils and spatial distribution of synaptic inputs to T4 and T5 cell arbors. (**A**) Retinotopic columnar terminals of Mi1 cells in stratum M1 (one center column and six surrounding columns). (**B**) The same subset of neurons in the proximal medulla (M10). Retinotopy of the columns is preserved in the distal lobula too. (**C**) The distribution of Tm2 axon terminals in layer Lo1. Locations of the columns are indicated by the broken circles. (**D**) Relative sizes and orientations of columnar units in different levels of the optic lobe. On the surface of the compound eye, ommatidial lenses are arranged in a regular hexagonal array. The arrangement of the lamina (LA) cartridges inherits the arrangement of the ommatidia, but is stretched in the D–V axis. The size of the column becomes larger in M1, where the lamina neurons make synapses with a number of medulla columnar neurons, including Mi and Tm cells. In M10, the size of the column becomes much smaller, and each column is elongated in the D–V axis. In the lobula, the column becomes yet smaller, and shortened in the A–P axis, making the column array triangular, rather than hexagonal. Center columns are in yellow, and six surrounding columns in black. (**E**) To normalize synaptic distributions on T4 and T5 dendrites, columns in M10 and Lo1 are assumed to be arranged in a regular hexagonal array, reflecting the arrangement of ommatidia. Input neurons to representative T4 and T5 cells were identified by the cell types and columns that they occupy, and their inputs were mapped onto the hexagonal grid by weighing synapses. The centers of

*Figure 5 continued on next page*

*Figure 5 continued*

the ellipses represent the averaged input sites, with the width and height corresponding to the standard deviations of inputs along the x- and y-axes. The left two columns are plotted using proofread synapses, whereas in the right columns, predicted (non-proofread) synapses were used. The average of all inputs was designated as the center point, which corresponds to the center of a column. The A–P and D–V axes correspond to the directions in the visual field, not to the body axis. The dimensions of the receptive fields are much more accurate, and are based on many more reconstructed neurons, than those reported from the manual annotation of ssTEM series in *Shinomiya et al. (2014)*. The synapses are also more comprehensively annotated, and the synapse annotation is much more precise. See the main text for further detail. Scale bar: (**A–C**) 5 μm.

DOI: https://doi.org/10.7554/eLife.40025.006

The following figure supplement is available for figure 5:

**Figure supplement 1.** Comparison between the numbers of predicted and proofread synapses per connection.

DOI: https://doi.org/10.7554/eLife.40025.007

output synapses of representative T4 and T5 cells manually, and corrected them as required. Neuron-to-neuron synaptic connections that were identified on the basis of manually annotated synapses correlated strongly with the identified connections based on auto-predicted synapses, indicating that neuronal connections of the pathways could initially be predicted without comprehensive labor-intensive manual synapse annotations. Comparing input neurons to T4 with those to T5 then enabled us to identify similarities in the cell types and the numbers of their synapses that underlie common computational strategies. The validity of such comparisons rests, of course, on the assumption that transmission at each anatomical synapse generates a unit postsynaptic signal of fixed size.

To assess the quality of the prediction, we counted numbers of synapses (both presynaptic and postsynaptic contacts) before and after manual proofreading of the T4 and T5 dendrites. For both counts, the numbers of output and input synapses before proofreading (predicted synapses) are plotted against the numbers of synapses after proofreading (proofread synapses) for the input/output synapses of T4 cell dendrites (*Figure 5—figure supplement 1A*), and T5 cell dendrites (*Figure 5—figure supplement 1B*). The numbers of synapses were counted for single T4 or T5 cells, whereas their synaptic partners were grouped together according to the cell type (e.g. Mi1, Tm1, CT1). For both T4 and T5, the numbers of predicted synapses almost always exceeded the numbers of proofread synapses, except for weak connections (<10–20 predicted synapses/connection), indicating that the predicted synapses include more false-positive (over-predicted) synapses than false-negative (under-predicted) synapses. The regression lines for both T4 and T5 cells largely fit each other, indicating that the quality of the synapse prediction is approximately constant between different subsets of neurons and different parts of the dataset (M10 and Lo1). This result therefore shows that the number of actual synapses can be reasonably estimated by calculating the ratio of false-positive or false-negative synapses, without exhaustive manual proofreading but by verifying only a small fraction of synapses manually.

On the basis of this result, the spatial distributions of input neurons to T4 and T5 cells were mapped using predicted rather than manually proofread synapses (*Figure 5E*, right two columns) and the same method as used for the left two columns. The results are consistent with the distribution patterns obtained using manually proofread synapses, although in some cases the input neurons failed to cluster clearly into three groups (e.g. T4d and T5c) because of a lack of spatial accuracy for the predicted synapses.

These analyses revealed that predicted synapses can be used directly for connectivity analysis without extensive manual proofreading, a great saving. In order to estimate the number of actual synapses from the predicted synapses, some of the predicted synapses still needed to be proofread to derive the ratio between the two values. In addition, we are aware that the spatial resolution is less reliable with predicted synapses than with manual proofreading, because the predicted synapses contain a number of false-positive or false-negative synapses, so that only strong neuron-to-neuron connections could be predicted with great accuracy, and that weak connections (about 10 true synapses/connection or fewer in this study; see *Figure 5—figure supplement 1*) may not be predictable with sufficient reliability.

## Discussion

### EMD circuit models based on actual connections

Classical correlation models of the motion detection circuit, including the *Hassenstein and Reichardt (1956)* and *Barlow and Levick (1965)* models, consider only two independent upstream inputs in the detection of motion. Several studies have provided physiological evidence that the EMD circuit may be approximated by either of these models or their modified versions (*Fisher et al., 2015*; *Gruntman et al., 2018*; *Leong et al., 2016*; *Salazar-Gatzimas et al., 2016*; *Strother et al., 2017*). These models cannot sufficiently address the asymmetrical responses of the T4 and T5 pathways, however, because the types and numbers of the neurons involved are limited, and are indeed not consistent with our findings that the input neurons to both T4 and T5 dendrites are clustered into three groups, not two (*Figure 5*). Previous studies, in particular, failed to include inhibitory inputs from CT1 to the T4 and T5 dendritic arbors. Although CT1 differs from the other medulla neurons providing inputs to T4 or T5 insofar as it lacks a direct synaptic partnership with lamina cells, it still receives indirect inputs from those cells via Mi1 and Mi9 (in M10) and Tm1 and Tm9 (in Lo1). CT1 is the only inhibitory columnar input to the T5 dendrites, and also the only element that is displaced from the other two excitatory legs. In the ON-edge side, CT1, together with the other two GABAergic neurons, C3 and Mi4, provides a measurable input to the base of the T4 dendrites.

As a foretaste to our anatomy, a new EMD circuit model with three parallel inputs was recently proposed, based on the two classical motion detection models, on computer simulations, and on activity recordings from T4 and T5 cells (*Haag et al., 2016*). Inputs to the motion-detecting unit include a non-delayed direct input, delayed enhancer input, and a delayed suppressor input (null direction suppression) located on the side opposite to the enhancer signal input, the two inputs bracketing the direct input (Figure 5a in *Haag et al., 2016*). The direction from the enhancer signal input to the suppressor signal input corresponds to the preferred direction (PD), and is opposite to the non-preferred direction (ND). This three-way input model incorporates the classical two-inputs models, the *Hassenstein and Reichardt (1956)* (H–R) and *Barlow and Levick (1965)* (B–L) models, as its subsets, so that the outputs, as well as the temporal tuning patterns of the circuits, are still consistent with the previous physiological studies. Applying this model to T4: a) Mi1 and Tm3, which provide inputs to the center of the dendritic arbor, would be direct inputs; b) Mi9 innervating the tip would be an enhancer input; and c) CT1, Mi4, and C3, which innervate the base, would be suppressor inputs. In the case of T5: a) Tm1, Tm2, and Tm4 would be direct inputs; b) Tm9 would be an enhancer, and; c) CT1 would be a suppressor input (*Figures 4*, *5* and *6*). Although Tm9 innervates the tip of T5 dendrites and therefore would fall into an enhancer location here, silencing experiments suggested that the contribution of Tm9 to the dark-edge detection was even larger than that of Tm1 and Tm2 combined (*Serbe et al., 2016*), so it might not be reasonable to regard Tm9 as an input that simply enhances the main direct inputs.

Among these inputs to the T-cells, CT1, C3, and Mi4 are known to be GABA-positive, Mi9 is glutamate-positive, and all other cells, including T4 and T5, are positive for cholinergic reagents (*Hasegawa et al., 2011*; *Kolodziejczyk et al., 2008*; *Pankova and Borst, 2017*; *Shinomiya et al., 2014*; *Strother et al., 2017*; *Takemura et al., 2017*). As all neurons providing inputs to the base are putatively GABAergic, and all neurons to the center putatively cholinergic, the connections of the ON- and OFF-edge pathways partially match the three-way EMD circuit model.

Both Mi9 and Tm9 cells relay signals from L3, and send inputs to the tip of the T4/T5 dendritic arbors. They both produce slow and sustained responses that serve as low-pass filters (*Arenz et al., 2017*; *Serbe et al., 2016*). Although Tm9 is putatively cholinergic and provides excitatory signals to T5 (*Fisher et al., 2015*; *Serbe et al., 2016*), putatively glutamatergic signals from Mi9 to T4 are supposed to be inhibitory on the basis of behavioral response assays (*Strother et al., 2017*). This possibility is also suggested by the observation that T4 expresses a glutamate-gated chloride channel, GluClα, which mediates inhibitory signals (*Strother et al., 2017*). Tm9 shows an increased response during OFF stimulation (*Arenz et al., 2017*; *Fisher et al., 2015*), and Mi9 is also thought to be activated similarly (*Arenz et al., 2017*; *Strother et al., 2017*) but provides input to the T4 pathway. These neurons may therefore modulate T4 and T5 using independent but opposing respective mechanisms. Mi9 may function as a temporal low-pass filter of the OFF signal (*Borst, 2018*) and may cancel noise during an OFF stimulus by suppressing T4, whereas Tm9 could enhance OFF signals to T5 along with the direct inputs from Tm1 and Tm2 (see panel (C) of *Figure 6—figure supplement*

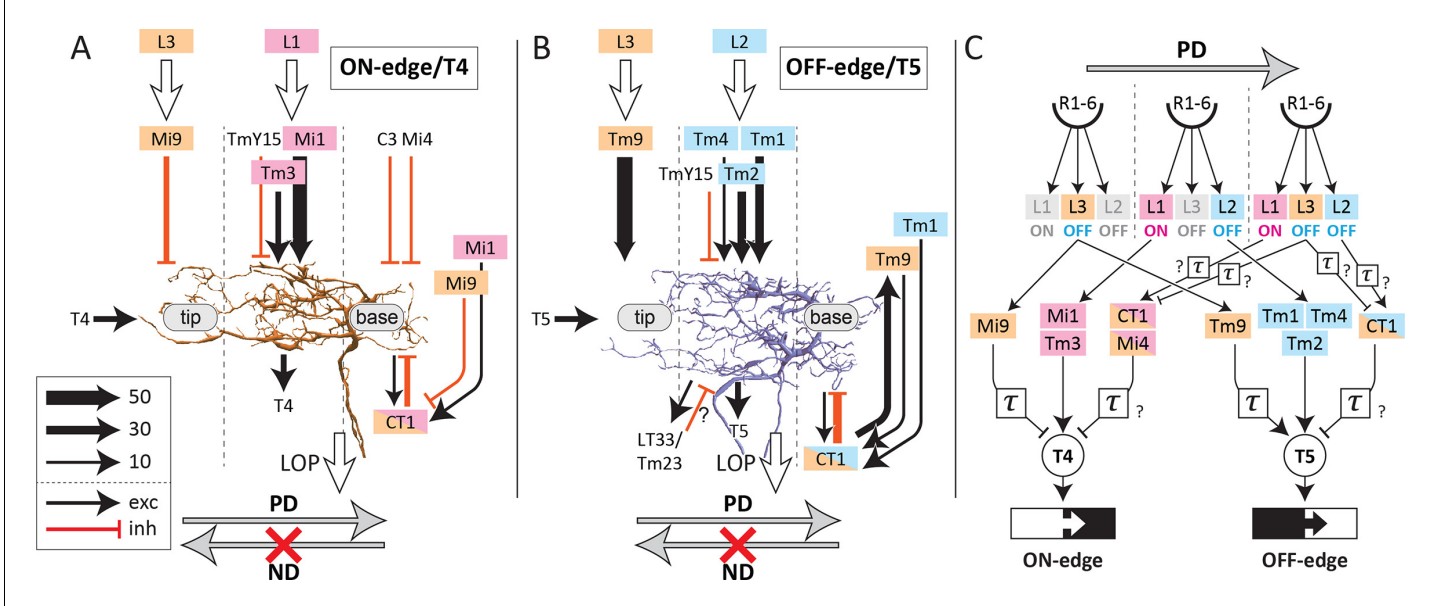

**Figure 6.** Schematic diagram of inputs onto and outputs from T4 and T5 dendrites. Major input and output neurons of the dendritic arbors of T4 (**A**) and T5 (**B**), ranked according to synapse number (key), with dotted lines indicating the approximate boundaries of the corresponding medulla/lobula columns. (**C**) Schematic circuit model of the ON-edge and OFF-edge pathways. A temporal delay is indicated by τ. See the text for details. Medulla neurons receiving inputs from L1, L2, and L3 cells are color-coded accordingly. PD, preferred direction; ND, non-preferred direction.

DOI: https://doi.org/10.7554/eLife.40025.008

The following figure supplement is available for figure 6:

**Figure supplement 1.** Circuit diagrams for ON- and OFF-edge models that signal motion.

DOI: https://doi.org/10.7554/eLife.40025.009

*1*). Besides Mi9 and Tm9, the tips of the T4 and T5 dendrites also receive excitatory inputs from T4 and T5 of the same cell type, a-d, which could also enhance signals to the EMD circuits by themselves, although these inputs are fewer in number and presumably would be far weaker than those from Mi9 or Tm9 (*Figure 2*).

CT1, which innervates the base of T4 and T5 dendrites, is an interesting wide-field cell that receives signals from the lamina cell pathways indirectly via other medulla neurons, including Mi1, Mi9, Tm1, and Tm9. The other inputs to the base of T4, Mi4 and C3, likewise lack direct inputs from lamina cells (*Takemura et al., 2017*), suggesting that inhibitory signals from CT1, Mi4, and C3 are delayed by an additional synapse relative to those from the direct inputs.

We can summarize the connectivity diagrams of ON- and OFF-edge EMD circuits that have now been demonstrated anatomically (*Figure 6*). A schematized and simplified EMD circuit diagram (*Figure 6C*), summarizes three downstream pathways from the photoreceptors (R1–R 6). Among the lamina cells, L1 signals to ON pathways selectively, whereas L2 and L3 both signal to OFF pathways (*Fisher et al., 2015*; *Strother et al., 2017*). None of the inputs to the base of T4 and T5, that is from CT1, Mi4, and C3, receives direct inputs from lamina interneurons. CT1 terminals in M10 receive only indirect input from L1 and L3, via Mi1 and Mi9, whereas those in Lo1 receive information from L2 and L3 only via Tm1 and Tm9 (*Figure 6A–B*). Mi4 also receives inputs from L1 and L3, through L5 and Mi9 (*Arenz et al., 2017*; *Takemura et al., 2017*). The transfer of information to these neurons may be delayed at additional synaptic relays (*Figure 6C*). Mi4, Mi9, and Tm9 themselves show delayed and sustained calcium responses against white-noise stimuli when compared with the responses of Mi1, Tm3, Tm1, Tm2, and Tm4, suggesting that the responses of Mi4, Mi9, and Tm9 serve as delayed arms in the EMD model (*Arenz et al., 2017*). The response properties of CT1 are still unknown. We suggest that the anatomical pathways of the two EMD circuits have now been reported in sufficient detail in this and previous accounts, but their physiological correlates, including the neurotransmitters and receptors of some constituent neurons, as well as temporal

delays at each cell, all still need further analysis to complete the picture of how the T-cells signal motion information.

On the basis of the neuronal connectivity depicted in *Figure 6*, speculative timing diagrams against ON- and OFF-edge signals to motion in the preferred and non-preferred directions are shown as *Figure 6—figure supplement 1*. Among the lamina cells, only L1 activates the down-stream cells during ON-edge signals. The direct inputs (Mi1 and Tm3) and suppressor inputs (CT1 and Mi4) to the T4 dendrites may therefore contribute to detecting ON-edge signals at the level of the EMD circuit. During OFF-edge signals, on the other hand, all three input legs to T5 as well as Mi9, which provides inhibitory inputs to T4, are activated. In both ON- and OFF-edge circuits, excitatory inputs provide signals to T4 or T5 cells first, before delayed inhibitory inputs suppress the responses of these cells to stimuli in the preferred direction. For signals in the non-preferred direction, T4 and T5 cells are inhibited by the suppressor inputs and will not be excited. Besides these, T4 is also likely to be suppressed by Mi9 not during ON-edge signals but during OFF-edge signals (*Strother et al., 2017*), presumably cutting off spontaneous noise from responses to non-preferred stimuli. Such a mechanism is lacking in the T5 pathway.

## Similarities and differences between the ON and OFF pathways

Inputs to T4 and T5 from their upstream neurons reveal significant anatomical similarities between the ON- and OFF-edge EMD circuits. The Mi1 cell in the ON pathway, for example, is similar to the Tm1 and Tm2 cells in the OFF pathway, insofar as all these cells provide inputs to the center of the T4/T5 cell dendrites and use acetylcholine as their neurotransmitter, and differ only in the neuropil of their termination. Like T4 (*Strother et al., 2017*), T5 expresses transcripts of two different nicotinic cholinoceptors, as well as those of an A-type muscarinic cholinoceptor, suggesting that T5 receives cholinergic Tm inputs by means of both ionotropic and metabotropic cholinoceptors (*Shinomiya et al., 2014*). Additional neurons, CT1 and TmY15, innervate both M10 and Lo1 and form synapses there with T4 and T5, respectively. On the other hand, three types of GABA-positive neurons provide inputs to the base of T4 dendrites, whereas only one type of GABAergic neuron (CT1) sends inputs to the base of T5. Inputs from CT1 make up a much larger proportion of the total input to T5 cells than to T4 cells (*Figure 2*). This difference might compensate for the lack of other inhibitory inputs to the base of T5. There are also two tangential elements in Lo1 (LT33 and Tm23) that make synapses with T5 dendrites, and counterparts are not found in M10. These then constitute differences betweeen the inputs to T5 and T4 cells. As the cell body site and projection trajectory of LT33 are both similar to those of CT1, a GABAergic cell, and as the cell is both presynaptic and postsynaptic to the T5 cell dendrites, it is possible that LT33 is also inhibitory and inhibits the activity of T5 regardless of the direction specificity. Even so, the T4 and T5 motion stimulus responses are very similar (*Haag et al., 2017*).

The T4 and T5 cells not only share functional characteristics, but also correlate closely in their development and are suggested evolutionary siblings. T4 and T5 are produced from the same lobula plate neuroblasts, and the expression of Notch specifies the generation of these two morphologically similar cell types (*Pinto-Teixeira et al., 2018*). Assuming that, during the course of evolution, T4 and T5 arose as duplicates from an ancestral cell population, as has been proposed (*Shinomiya et al., 2015*), the neurons that provide their synaptic partnerships, most notably Mi and Tm cells, could also have been duplicated, possibly in an event that was induced by the duplication of T4/T5 (*Fischbach and Dittrich, 1989*; *Shinomiya et al., 2015*).

T4 and T5, or morphologically similar optic lobe neurons, have been found in a broad range of arthropod species, including various Diptera, the honeybee, butterfly, and crab (*Bengochea et al., 2018*; *Buschbeck and Strausfeld, 1996*; *Cajal and Sánchez, 1915*; *Strausfeld, 2012*), suggesting that the origin of these cells can be traced back to the Cambrian, when pancrustacean ancestors are thought to have given rise to hexapod and crustacean species (*Rota-Stabelli et al., 2013*). Anatomical and functional differences in the T4 and T5 pathways of the fly's brain may therefore have accumulated during the course of evolution, adapting ancestral forms to their living environments, by changes in the synaptic connections of their partner neurons.

## Possible roles for CT1 and TmY15

The two novel cells described in our report, CT1 and TmY15, share important similarities: both are putatively inhibitory, and both provide input to both T4 and T5. They differ chiefly in their anatomical field size. TmY15 is narrow-field, spanning at least 10 columns, whereas CT1 spans the entire field. The architecture of the connecting networks also differs. Thus, both are anatomically qualified to provide an inhibitory surround to the field of T-cells that they innervate.

In addition to its similarity to TmY15, CT1 may support a local computation of the inhibitory elements of a Barlow-Levick circuit. Calculations of the space constant for the CT1 arbor suggest that a delay between adjacent columns is sufficient to allow local inhibition, in addition to any global inhibition that this cell may mediate. Such local computation has both the right sign and location (on the leading edge of the anti-preferred direction) in both the ON-pathway of T4 in the medulla, where CT1 is excited by Mi1 and Tm3 (*Takemura et al., 2013*), and the OFF-pathway of T5 in the lobula, where the inputs are instead Tm1 and Tm9. The latter, in turn, derive their inputs from L1 (to T4) and L2 (to T5).

The functional significance of TmY15 in motion processing must remain speculative. Not only does this cell receive a wide range of inputs from various types of cells in the medulla, lobula, and lobula plate, but it also has much weaker synaptic contacts with T4 and T5 in M10 and Lo1 when compared with other Mi or Tm cell inputs. It specifically innervates the second and third strata (Lop2 and Lop3) in the lobula plate, and preliminary observations show that it receives inputs from cells T4b, T4c, T5b, and T5c, suggesting that TmY15 may also work as a feedback loop to suppress responses in T4 and T5 during regressive and upward motions.

In summary, our report comprehensively identifies input and output neurons of the dendritic arbors of T4 and T5 cells, and uses a single dataset to reveal (at synapse level) the detailed similarities between the connections of these two motion-signaling output cells. Together with the functional contribution of individual neurons in the motion-detection circuits shown in several studies (*Behnia et al., 2014*; *Fisher et al., 2015*; *Meier et al., 2014*; *Serbe et al., 2016*), the detailed connectivity diagram that we provide here should further facilitate functional analyses in these cells, through behavioral assays, calcium imaging, and electrophysiological recordings, and by providing comparisons with known neurons in the ON-pathway.

## Materials and methods

### Sample preparation, imaging, and image processing

We first dissected the right eye and optic lobe of a 6-day post-eclosion female fruit fly, *Drosophila melanogaster*, a cross between homozygous $w^{1118}$ and CS wild type (*Takemura et al., 2013*). For this, the fly's head was dissected to reveal the brain, which was prefixed for 20 min in 2.5% each of paraformaldehyde and glutaraldehyde in 0.1M cacodylate buffer. Then, 170 µm thick Vibratome slices were high-pressure frozen and freeze-substituted in 1% $OsO_4$, 0.2% uranyl acetate, 3% water in acetone, before embeddeding in Durcupan, all as previously reported (*Takemura et al., 2015*; *Takemura et al., 2017*).

### FIB-SEM dataset

Adopting previous methods (*Knott et al., 2008*; *Xu et al., 2017*), we used a Zeiss NVision to combine four successive 8 × 8 nm FIB-SEM images each at 2 nm depth, to yield a stack of 8 nm voxel cubes covering the central part of the medulla, lobula, and lobula plate, together with the second optic chiasm connecting these neuropils. The size of the image stack was 19,162 × 10,657 × 22,543 pixels, equivalent to 153 µm x 85 µm x 180 µm. To identify all T4/T5 inputs, we needed first to segment the FIB-SEM data volume three-dimensionally using an algorithm that recognizes cell membranes, and to reconstruct neurons manually by merging segmented fragments of neurons through a process of 'proofreading' (*Takemura et al., 2015*). The optic lobe FIB-SEM dataset used in this study is available at http://emdata.janelia.org/optic-lobe/.

### Automated segmentation and synapse prediction

The dataset was trained and classified using a variant of the context-aware agglomeration workflow described in *Parag et al., 2015*. Several modifications to this pipeline improved the segmentation of

neurons having membranes that were broken during sample fixation. These modifications included: 1) training labels added to identify the location of broken membranes, which augments the multi-class (boundary, cytoplasm, mitochondria) prediction already used; 2) voxel prediction using the auto-context workflow in ilastik (*Sommer et al., 2011*); and 3) an initial watershed based on the distance transform (*Beier et al., 2017*). This pipeline was implemented and run using the pipeline described in *Plaza and Berg (2016)*.

The prediction of putative synaptic sites in the dataset was performed according to *Huang et al. (2018)*. Our final dataset comprised >1,750,000 presynaptic sites identified by specific organelles, T-bar ribbons (*Takemura et al., 2008*), opposite which were neurites with >11,500,000 postsynaptic densities (PSDs). To produce synapse predictions, a deep-learning convolutional network was first used to predict presynaptic sites (*Huang et al., 2016*; *Huang et al., 2018*). Then, conditioned on the output of the automatic segmentation, a multilayer perceptron was used to predict partnering PSDs for each predicted presynaptic site. These automated systems were trained from manual annotations given on small subvolumes of $(500)^3$ pixels.

### Neuron tracing and synapse proofreading

T4 and T5 cells were selected in the auto-segmented dataset from their distinctive arbors in M10 and Lo1, respectively, and entire neurons were reconstructed using EM tracing and visualization software (NeuTu: https://github.com/janelia-flyem/NeuTu; *Feng et al., 2015*). T4 and T5 subtypes a–d project their terminals to consecutive strata Lop1–Lop4 in the lobula plate (*Fischbach and Dittrich, 1989*), and for each subtype, five representative neurons were identified and fully traced. Using the predicted synapses as a template, all synaptic sites of these neurons were verified manually and corrected if they failed to match pre- and postsynaptic sites. After the synapses were exhaustively annotated, all neurites that were either pre- or postsynaptic to the representative T4 or T5 cell subtypes were further traced and their cell types identified, on the basis of their morphology and connections.

### Genetics, immunohistochemistry and light microscopy

To visualize LT33 and Tm23 cells by light microscopy, we used MultiColor FlpOut (MCFO) (*Nern et al., 2015*), a genetic method for brainbow-like labeling of individual cells in multiple colors. Split-GAL4 driver lines with expression in LT33 and Tm23 were crossed to MCFO-1 (Tm23) or MCFO-7 (LT33) and processed for immunolabeling, mounted in DPX and imaged as described before (*Nern et al., 2015*). Anti-Bruchpilot (Brp) (labeled using mAb nc82, obtained from the Developmental Studies Hybridoma Bank (*Wagh et al., 2006*), was included as a reference marker as described before (*Nern et al., 2015*). The driver lines, SS01018 (R23H11-AD; R83H07-DBD) (LT33) and SS02415 (VT025526-AD, VT044322-DBD) (Tm23), both have expression in additional cell types but were sufficiently specific to allow us to obtain multiple examples of MCFO-labeled cells for each type. Split-GAL4 hemidrivers and parent GAL4 lines have been described before (*Dionne et al., 2018*; *Jenett et al., 2012*; *Tirian and Dickson, 2017*). To show cell shapes in different orientations, the resampled views shown in *Figure 3F, H, and I* were generated from image stacks using Vaa3D (Neuron-Annotator mode) (*Peng et al., 2010*) and exported as TIFF screenshots. *Figure 3F and I* show segmented images that were edited to exclude additional labeled cells (distinguished by labeling color or location).

## Acknowledgements

The authors thank Drs Michael Reiser, Shin-ya Takemura, and Eyal Gruntman for fruitful discussions and their critical comments on the project. For data processing and management, we would also like to acknowledge Lowell Umayam, Donald Olbris, William Katz, Stuart Berg and Charlotte Weaver in the FlyEM Project Team of the Janelia Research Campus. This study was funded and supported by the Howard Hughes Medical Institute, and supported by a Kazato Research Encouragement Prize from the Kazato Research Foundation, to KS.

## Additional information

### Funding

| Funder | Grant reference number | Author |
| --- | --- | --- |
| Howard Hughes Medical Institute | | Stephen M Plaza |
| Kazato Research Foundation | Kazato Research Encouragement Prize | Kazunori Shinomiya |

The funders had no role in study design, data collection and interpretation, or the decision to submit the work for publication.

### Author contributions

Kazunori Shinomiya, Conceptualization, Data curation, Formal analysis, Funding acquisition, Validation, Investigation, Visualization, Methodology, Writing—original draft, Writing—review and editing; Gary Huang, Toufiq Parag, Resources, Software, Methodology; Zhiyuan Lu, C Shan Xu, Resources; Roxanne Aniceto, Namra Ansari, Natasha Cheatham, Shirley Lauchie, Erika Neace, Omotara Ogundeyi, Christopher Ordish, David Peel, Aya Shinomiya, Claire Smith, Satoko Takemura, Iris Talebi, Data curation; Patricia K Rivlin, Project administration; Aljoscha Nern, Resources, Visualization, Writing—review and editing; Louis K Scheffer, Formal analysis, Supervision, Writing—review and editing; Stephen M Plaza, Conceptualization, Supervision, Funding acquisition, Methodology, Project administration, Writing—review and editing; Ian A Meinertzhagen, Conceptualization, Supervision, Writing—original draft, Writing—review and editing

### Author ORCIDs

Kazunori Shinomiya (iD) http://orcid.org/0000-0003-0262-6421
C Shan Xu (iD) http://orcid.org/0000-0002-8564-7836
Aljoscha Nern (iD) https://orcid.org/0000-0002-3822-489X
Louis K Scheffer (iD) https://orcid.org/0000-0002-3289-6564
Ian A Meinertzhagen (iD) https://orcid.org/0000-0002-6578-4526

### Decision letter and Author response

Decision letter https://doi.org/10.7554/eLife.40025.014
Author response https://doi.org/10.7554/eLife.40025.015

## Additional files

### Supplementary files

• Transparent reporting form
DOI: https://doi.org/10.7554/eLife.40025.010

### Data availability

The raw electron microscopy data, as well as skeletonized neurons and numbers of input and output synapses of T4 and T5 cells used in this study are hosted at a Janelia website: http://emdata.janelia.org/optic-lobe/.

The following dataset was generated:

| Author(s) | Year | Dataset title | Dataset URL | Database and Identifier |
| --- | --- | --- | --- | --- |
| Stephen M Plaza | 2019 | fib19-grayscale | http://emdata.janelia.org/optic-lobe/ | Janelia, fib19-grayscale |

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
