## [Decision Letter]

Thank you for submitting your article "Comparative connectivity analysis of the ON- and OFF-edge motion pathways in the *Drosophila* brain" for consideration by *eLife*. Your article has been reviewed by three peer reviewers, including Alexander Borst as the Reviewing Editor and Reviewer #1, and the evaluation has been overseen by K VijayRaghavan as the Senior Editor.

The reviewers have discussed the reviews with one another and the Reviewing Editor has drafted this decision to help you prepare a revised submission.

Summary:

The authors used a focused ion beam scanning electron microscope (FIB-SEM) to acquire high resolution images of a large fraction of a *Drosophila* optic lobe. The dataset is superior to previous EM datasets because it is of higher isotropic resolution and encompasses the complete pathway from the lamina to the lobula plate. From this dataset they reconstructed five T4 and T5 neurons – elementary motion detectors in the fly visual system – detected all their synaptic contacts (pre- and postsynaptic, with the exception of the lobula plate) and traced and identified the connected cell types. While the circuitry in the ON-selective pathway had already been described in detail previously, the authors were able to demonstrate for the first time the spatial distribution of the presynaptic elements to OFF-selective T5 cells, namely inputs segregated into distal, central, and basal dendritic inputs. In addition to confirming already published findings (columnar cell types Tm9, Tm2, Tm1, and Tm4 as major inputs to T5), they describe a new potential key-player in the T5-circuit, CT1.

The data presented here by Shinomiya and colleagues, especially the new evidence for the spatial distribution of T5 inputs, is convincing and with no doubt of great interest for the scientific community. In particular, the manuscript aims at constraining the possible circuits for direction selectivity in the *Drosophila* visual system with EM data, and, thus, is a timely and interesting addition to this field.

Essential revisions:

General criticism:

The quality of the writing as well as the structure of the manuscript are very unsatisfying. Overall poor referencing. The figures are poorly linked to the text. The order of figures and the Results part of the manuscript do not follow a consistent logic.

Section-by-section criticism:

A) Title: I would propose to focus on the new finding, and not on the comparison with the T4 inputs. I would propose the following title: Connectivity analysis of the *Drosophila* OFF-edge motion pathway.

B) Introduction: There are some abrupt jumps from modeling (first section) to anatomy without introducing the optic lobe structure at all, pointing out the roles of the described elements in the context of the motion vision circuitry. I would propose to restructure the introduction: 1) Models for motion detection. 2) General architecture of the optic lobe. 3) Functional evidence; split ON v. OFF (L1,L2; T4,T5). 4) Anatomical evidence already published (presynaptic elements to T4 and T5). 5) Functional evidence.

C) Results: The figure referencing in the Results part does not follow any logic. Every part of every figure should be referenced in the text before the next one. I would suggest to rearrange the figures in the beginning of the Results: 1) Change the order of Figures 2 and 3. 2) First describe the inputs to T4 and T5 and then their anatomy. 3) Move Figure 6 to supplements. 4) The last two sections (about CT1 and TmY15) would better fit to the corresponding section where anatomy and connectivity is described.

D) Discussion: The second paragraph of the subsection “Three-way EMD circuit model” is largely confusing and does not contribute to the understanding of the subject. I would suggest removing this section from the manuscript. The Discussion would profit from referencing the corresponding figures more regularly.

Another concern is that the interpretation of the connectomics data is exclusively done through the lens of the three-arm model from Haag et al., 2016, which incorporates both Hassenstein-Reichardt style preferred direction enhancement and Barlow-Levick style null direction suppression. This is an important model but there is, however, no consensus at the moment on how direction selective signals emerge, with studies reporting both/either preferred direction enhancement and/or null direction suppression (Fisher et al., 2015; Salazar-Gatzimas et al., 2016; Leong et al., 2016; Strother et al., 2017; Gruntman et al., 2018). While it is tempting to focus on the three-arm model, because of their and previous findings that the inputs are separated into three clusters, it also seems overly restrictive. It is important to mention alternative models, and how this connectomics data might inform these.

Loosely related to this point, there is no mention of silencing experiments performed by various groups (Strother et al., 2017; Ammer et al., 2015; Serbe et al., 2014; Silies et al., 2013). Even though these are not always conclusive, they are directly relevant to the final model put forward in Figure 7.

[Editors' note: further revisions were requested prior to acceptance, as described below.]

Thank you for resubmitting your work entitled "Comparisons between the ON- and OFF-edge motion pathways in the *Drosophila* brain" for further consideration at *eLife*. Your revised article has been favorably evaluated by K VijayRaghavan as the Senior Editor, and a Reviewing Editor.

The revised manuscript has improved substantially, both in its structure as well as in the writing. However, before it can go to press, the authors should fix the following things:

1) Writing:

Some sentences are still too long and need to be shortened. In general, the manuscript would profit from one of the authors smoothing out the language.

The following sentences from the Introduction might serve as an example: “Functionally, both T4 and T5 cells are direction-selective, and each is further grouped into four subtypes: T4 as T4a, T4b, T4c and T4d, and T5 as T5a, T5b, T5c, and T5d. T4 and T5 cells specifically extract motion in the four canonical directions. The subtypes a-d, detect front-to-back, back-to-front, upward, and downward motion, respectively one each (Maisak et al., 2013).”

In a similar way, the last paragraph of the Introduction justifies the present study because previous data sets used different methods and are each lacking specific parts of the optic lobe. But then it diffuses again in details like: “The latter are paired Tm1 and Tm2 cells with multiple Tm4 cells that all receive L2 input, and a fourth cell Tm9 (Shinomiya et al., 2014), a medulla target of L3, the third lamina neuron in each cartridge. Thus Tm1 and Tm2 provide input from a single column while Tm4 cells provide input originating from multiple columns. Tm9 in turn provides a candidate substrate for L3's combinatorial action with L2 (Silies et al., 2013).” I would simply delete these sentences, focusing on the main point made above.

2) Other major point:

In the third paragraph of the subsection “Identifying the synaptic partners of T4 and T5 cells2”, it states Tm1, Tm4, and Tm9 as CT1 input. The first and sixth paragraphs of the subsection “EMD circuit models based on actual connections”, state only Tm1 and Tm9, while in the second paragraph of the subsection “Possible roles for CT1 and TmY15”, the authors write that Tm1 and Tm4 are presynaptic to CT1. In Figure 6 only Tm1 and Tm9 are depicted as CT1 inputs. Please also clarify where these data come from and how strong these inputs are (synapse numbers).

---

## [Author Response]

Essential revisions:General criticism:The quality of the writing as well as the structure of the manuscript are very unsatisfying. Overall poor referencing. The figures are poorly linked to the text. The order of figures and the Results part of the manuscript do not follow a consistent logic.Section-by-section criticism:A) Title: I would propose to focus on the new finding, and not on the comparison with the T4 inputs. I would propose the following title: Connectivity analysis of the Drosophila OFF-edge motion pathway.

We agree with the suggestion that the title should focus on the new findings, i.e. on the OFF-edge pathway. A major merit of our work is, however, we suggest, that both pathways, those to T5 (which is new) and to T4 (which is an extended update) have now been made from the same FIB-SEM dataset using identical methods without biases, and discuss motion processing models by referring to previous works on both ON and OFF pathways. These use some of the same pathways. So, while we agree to the need to underemphasize the T4 pathway, we think there is still merit in referring to its comparison. We therefore now use the following title: “Comparisons between the ON- and OFF-edge motion pathways in the *Drosophila* brain.”

B) Introduction: There are some abrupt jumps from modeling (first section) to anatomy without introducing the optic lobe structure at all, pointing out the roles of the described elements in the context of the motion vision circuitry. I would propose to restructure the introduction: 1) Models for motion detection. 2) General architecture of the optic lobe. 3) Functional evidence; split ON v. OFF (L1,L2; T4,T5). 4) Anatomical evidence already published (presynaptic elements to T4 and T5). 5) Functional evidence.

We agree and have significantly revised the entire Introduction. We have followed the reviewer’s suggested ordering of topics starting with models for motion detection. The general architecture of the optic lobe is now covered by new text in the second paragraph of the Introduction. We then follow this by reporting the division into ON and OFF pathways, in the fourth paragraph of the Introduction.

C) Results: The figure referencing in the Results part does not follow any logic. Every part of every figure should be referenced in the text before the next one. I would suggest to rearrange the figures in the beginning of the Results: 1) Change the order of Figures 2 and 3. 2) First describe the inputs to T4 and T5 and then their anatomy.

We have followed these suggestions from the reviewer and now reverse Figures 2 and 3 from their order in the previous text. We are careful to cite the figures and figure panels in order.

3) Move Figure 6 to supplements.

We have done this. The figure is now Figure 5—figure supplement 1.

4) The last two sections (about CT1 and TmY15) would better fit to the corresponding section where anatomy and connectivity is described.

We agree with the reviewer’s suggestion and the description for CT1 and TmY15 are now moved to the anatomy section (subsection “Identifying the synaptic partners of T4 and T5 cells”).

D) Discussion: The second paragraph of the subsection “Three-way EMD circuit model” is largely confusing and does not contribute to the understanding of the subject. I would suggest removing this section from the manuscript. The Discussion would profit from referencing the corresponding figures more regularly.

We now realize that part of our original text was incorrect, and we have dealt with this. We think this helps to remedy the reviewer’s confusion. We think it important to compare the responses of Mi9 and Tm9, moreover, and therefore decide to retain the edited version of this paragraph.

Another concern is that the interpretation of the connectomics data is exclusively done through the lens of the three-arm model from Haag et al., 2016, which incorporates both Hassenstein-Reichardt style preferred direction enhancement and Barlow-Levick style null direction suppression. This is an important model but there is, however, no consensus at the moment on how direction selective signals emerge, with studies reporting both/either preferred direction enhancement and/or null direction suppression (Fisher et al., 2015; Salazar-Gatzimas et al., 2016; Leong et al., 2016; Strother et al., 2017; Gruntman et al., 2018). While it is tempting to focus on the three-arm model, because of their and previous findings that the inputs are separated into three clusters, it also seems overly restrictive. It is important to mention alternative models, and how this connectomics data might inform these.

We absolutely agree, and in the first paragraph of the Discussion, we have therefore cited the additional references (Fisher, Salazar-Gatzimas, Leong, Strother, Gruntman) mentioned by the reviewer although we do not think that these models fit our anatomical observations as clearly as the three-arm model of Haag et al., 2016.

Loosely related to this point, there is no mention of silencing experiments performed by various groups (Strother et al., 2017; Ammer et al., 2015; Serbe et al., 2014; Silies et al., 2013). Even though these are not always conclusive, they are directly relevant to the final model put forward in Figure 7.

The studies mentioned by the reviewer refer to both ON and OFF pathways. They are now cited and provide an additional reason to retain reference to both pathways in the title to our report (see above, comment under “Title”).

[Editors' note: further revisions were requested prior to acceptance, as described below.]

The revised manuscript has improved substantially, both in its structure as well as in the writing. However, before it can go to press, the authors should fix the following things: 1) Writing: Some sentences are still too long and need to be shortened. In general, the manuscript would profit from one of the authors smoothening the language. The following sentences from the Introduction might serve as an example: “Functionally, both T4 and T5 cells are direction-selective, and each is further grouped into four subtypes: T4 as T4a, T4b, T4c and T4d, and T5 as T5a, T5b, T5c, and T5d. T4 and T5 cells specifically extract motion in the four canonical directions. The subtypes a-d, detect front-to-back, back-to-front, upward, and downward motion, respectively one each (Maisak et al., 2013).”In a similar way, the last paragraph of the Introduction justifies the present study because previous data sets used different methods and are each lacking specific parts of the optic lobe. But then it diffuses again in details like: “The latter are paired Tm1 and Tm2 cells with multiple Tm4 cells that all receive L2 input, and a fourth cell Tm9 (Shinomiya et al., 2014), a medulla target of L3, the third lamina neuron in each cartridge. Thus Tm1 and Tm2 provide input from a single column while Tm4 cells provide input originating from multiple columns. Tm9 in turn provides a candidate substrate for L3's combinatorial action with L2 (Silies et al., 2013).” I would simply delete these sentences, focusing on the main point made above.

We carefully rechecked the manuscript and did our best to streamline sentences. We hope the entire article now reads smoothly.

We admit that the part (“The latter are paired Tm1 and Tm2 cells …”) was off the topic and also redundant. We therefore now removed this part from the text.

2) Other major point:In the third paragraph of the subsection “Identifying the synaptic partners of T4 and T5 cells2”, it states Tm1, Tm4, and Tm9 as CT1 input. The first and sixth paragraphs of the subsection “EMD circuit models based on actual connections”, state only Tm1 and Tm9, while in the second paragraph of the subsection “Possible roles for CT1 and TmY15”, the authors write that Tm1 and Tm4 are presynaptic to CT1. In Figure 6 only Tm1 and Tm9 are depicted as CT1 inputs. Please also clarify where these data come from and how strong these inputs are (synapse numbers).

Including Tm4 as a major CT1 input was our mistake. While both CT1 receives inputs from both Tm9 and Tm1 with substantial numbers of synapses (23.3 ± 2.6 and 7.0 ± 3.1 synapses per CT1 terminal, respectively (n=7)), Tm4 has virtually no synapses with CT1 (0.4 ± 0.8 synapses per CT1 terminal (n=7)). So we have removed Tm4 from these statements and added synapse numbers for Tm9 and Tm1. We apologize for the confusion.